# What We Have Learned from Animal Models to Understand the Etiology and Pathology of Endometrioma-Related Infertility

**DOI:** 10.3390/biomedicines10071483

**Published:** 2022-06-23

**Authors:** Zhouyurong Tan, Sze-Wan Hung, Xu Zheng, Chi-Chiu Wang, Jacqueline Pui-Wah Chung, Tao Zhang

**Affiliations:** 1Department of Obstetrics and Gynaecology, The Chinese University of Hong Kong, Hong Kong; zyrtan@link.cuhk.edu.hk (Z.T.); szewanhung@cuhk.edu.hk (S.-W.H.); zhengxu@link.cuhk.edu.hk (X.Z.); ccwang@cuhk.edu.hk (C.-C.W.); jacquelinechung@cuhk.edu.hk (J.P.-W.C.); 2Reproduction and Development, Li Ka Shing Institute of Health Sciences, The Chinese University of Hong Kong, Hong Kong; 3School of Biomedical Sciences, The Chinese University of Hong Kong, Hong Kong; 4Sichuan University-Chinese University of Hong Kong Joint Laboratory in Reproductive Medicine, The Chinese University of Hong Kong, Hong Kong

**Keywords:** endometrioma, infertility, models, pathophysiology, therapeutic targets

## Abstract

Endometrioma (OMA) is the most common subtype of endometriosis, in which the endometriotic lesions are implanted in the ovary. Women with OMA are usually associated with infertility, presenting with reduced ovarian reserve, low oocyte quantity and quality, and poor fertility outcomes. However, the underlying pathological mechanisms in OMA-related infertility are still unclear. Due to the limitations and ethical issues of human studies in reproduction, animal models that recapitulate OMA characteristics and its related infertility are critical for mechanistic studies and subsequent drug development, preclinical testing, and clinical trials. This review summarized the investigations of OMA-related infertility based on previous and latest endometrioma models, providing the possible pathogenesis and potential therapeutic targets for further studies.

## 1. Introduction

Endometrioma (OMA) occurs within ovaries and manifests as single or multiple distinct cysts that are also vividly named chocolate cysts because of their brown, tar-like appearance, similar to melting chocolate [1]. It is the most common subtype, affecting up to 40% of patients with endometriosis [2]. There are other two subtypes of endometriosis: (1) superficial peritoneal endometriosis (SUP) and (2) deep infiltrating endometriosis (DIE). The former lies on the lining of the peritoneum and the latter is characterized by lesions infiltrating over 5 mm under the peritoneal surface [3]. On account of heterogeneity in location, these three forms of endometriosis present vast variations in color, size, and invasion depth, which may contribute to the disparity in related symptoms and severity of the diseases [4,5]. OMA always manifests in moderate or severe stages, is associated with infertility, ovarian or pelvic adhesions, and the risk of ovarian cancer [6,7,8,9,10].

A positive correlation between endometriosis and infertility has been shown in several publications [9,11,12,13,14,15]. However, the pathophysiology of impaired fertility in patients with endometriosis remains unclear [11]. From current studies, several possible causes may contribute to the reduced fecundity in women with endometriosis, depending on endometriosis subtypes [16]. When endometriosis extends to the ovaries to form OMA, it is speculated to damage functional ovarian tissue via space-occupying effects and local reactions. This results in reduced ovarian reserve as assessed by ultrasound and follicle-stimulating hormone levels [17]. Although there are various possible mechanisms for reduced fertility in women with OMA, up to now, the primary cause has yet to be identified.

Current therapeutic options for endometriosis usually aim to ameliorate symptoms i.e., pain relief [18]. Hung et al. have well summarized the current medical treatments for endometriosis [19]. Thereinto, the first-line agents including combined oral contraceptives (COC) and progestin are under the mechanism of reducing estrogen effects and suppressing ovulation. These drugs significantly reduced the recurrence of dysmenorrhea [20]. While their application leads to the risk of impaired infertility, symptoms related to hypoestrogenism, and thromboembolism [19]. Gonadotropin-releasing hormone (GnRH) agonist is the second-line therapy of endometriosis to ameliorate the symptoms and prevent recurrence after surgery by suppressing ovarian production of estrogen and creating a hypoestrogenic state. Similarly, it also brings long-term side effects such as osteoporosis because of the increased bone turnover and subsequent reduction in bone mineral density [19,21]. Therefore, the current medicines may not be suitable for patients with OMA-associated infertility who have to conceive desire [22]. Surgical treatments including laparoscopy and laparotomy are widely used for the diagnosis or lesion resection [23]. However, it is still under debate whether surgical treatment of OMA removal will benefit pregnancy outcomes. On one hand, a meta-analysis analyzed 13 studies and reported that ovarian reserve assessed by antral follicle count (AFC) was not reduced with surgical treatment of OMA [24]. On the other hand, a systematic review and prospective cohort studies reported a decreased ovarian reserve as the consequence of surgical excision of OMA [25,26,27]. The discrepancy may come from variations in surgical skills between surgeons and different stages of endometriotic lesions. Instead of the possibility of ovarian injury [28], surgical interventions may lead to other potential side effects, such as postoperative adhesions, surgical complications, and delayed infertility treatment [29], yet, there was also a high possibility of endometriosis recurrence in second-year post-operation [30]. To maximize and restore sub-fertility in OMA treatment, artificial reproduction technology (ART) was proposed to apply solely or combined with surgery [29,30,31,32]. A systematic review reported that during sole application of ART, patients with OMA had a higher cycle cancellation rate and lower oocyte yield [33]. The option of OMA surgery could be considered before ART, while whether the surgical removal of OMA benefits ART result remains risks. It was indicated that women with surgical removed OMA, compared with untreated women with OMA, had comparable oocyte retrieval rate, clinical pregnancy rate, and live birth rate [33]. Moreover, according to the ESHRE guideline, there was no evidence to show that the removal of OMA lesion larger than 3 cm before ART could improve pregnancy rates [34]. In addition, ART is not cost-effective, which restricts its availability and affordability [35].

It has no doubt that both medical treatment and surgical intervention are not satisfactory. Further research on the therapeutic targets and novel treatments for OMA-related infertility is urgently needed. Nevertheless, it raises ethical concerns to study OMA-related infertility in humans due to the limited sample access. As well, the clinical trials in humans have been strictly controlled especially on infertility treatment [36]. Therefore, different animal models have been established to recapitulate the key features of the OMA condition, which aids to understand the impact and etiology of OMA on fecundity parameters. Here our review aims to summarize how different animal models could help to study therapeutic agents for inhibiting OMA development and to improve fertility outcomes. We also proposed the need of further establishment of a more proper OMA-related infertility model and suggest the potential treatment for enhancing the pregnancy outcome.

## 2. Methodology

A search was performed on Pubmed using the search terms ((“Endometriosis”[Mesh]) AND ((“Fertility”[Mesh]) OR (“Infertility”[Mesh]) OR (“Reproduction”[Mesh]))) AND (“Models, Animal”[Mesh]) on 5 February 2022. All titles and abstracts were screened to verify the mentioned animal model of endometriosis-related infertility and 128 articles were screened from the beginning. Thirty-nine related articles were extracted from other reviews. After screening out reviews, irrelevant and duplicate articles, 82 articles were identified. Thereinto, 3 articles indicated the animal models for OMA in endometriosis-associated infertility.

## 3. Animal Models of Endometriosis-Associated Infertility and OMA-Related Infertility

Currently, the mainstream study of the mechanisms underlying endometriosis-associated infertility is based on limited human samples, such as tissue fragments, primary cells, cell lines, and fluids (i.e., peritoneal fluids, endometriotic fluids, follicle fluids), obtained through surgery or assisted reproductive technology (ART) [37]. Although some mechanisms (i.e., adhesion, hormonal milieu, inflammation, and immunology) are regarded as possible pathophysiology of endometriosis-associated infertility [38], the intertwined relationships between various pathophysiological disturbances are also highlighted [8]. In vitro studies fail to mimic the in vivo microenvironments including cell-cell/cell-extracellular matrix communication, as well as cellular diversity in the natural cell environment. Therefore, in vivo experimental models that simulate the main features of diseases and microenvironments are emergently needed. In recent 40 years, animal models of endometriosis had been widely developed [39]. Nevertheless, most of these models mimic subtypes of SUP and DIE, and few mimic OMA. The lack of specific in vivo models which resemble characteristics of OMA and its related infertility may impede the progress of these studies. This section provides in-depth investigations of the methods of the existed endometriosis and OMA models which were applied to study the associated infertility. 

### 3.1. Endometriosis Models and Infertility

#### 3.1.1. Non-Human Primates

The most widely accepted pathophysiology of endometriosis is retrograde menstruation, proposed by Sampson in 1921, in which endometriosis arises from endometrial cells/debris shedding from the uterus via the fallopian tubes into the ovary or other sites in the pelvic cavity [40,41,42]. Non-human primates (NHPs) like baboons and cynomolgus monkeys have absolute advantages as they are phylogenetically proximate to humans, with similarities in aspects of anatomy, physiology, as well as pathology [43]. NHPs have cyclic menstruation and can develop endometriosis spontaneously. Therefore, they are extensively used as experimental endometriosis animal models to study its associated infertility [44]. Endometriosis symptoms in cynomolgus monkeys are similar to those in women [45,46]. In 1992, Ami et al. demonstrated that the majority of cynomolgus monkeys had lesion cysts and mostly were accompanied by uterine adenomyosis [45]. Implications of sub-infertility in the NHP model of endometriosis are widely investigated to evaluate the disease progression and to monitor drug efficacy [41].

However, the shortcomings of the endometriosis model in NHPs cannot be ignored. Firstly, NHPs develop endometriosis at a lower rate than human beings and cannot be diagnosed until the related signs and symptoms develop severely [47,48]. Many researchers tried other methods to induce endometriotic lesions artificially in NHP. Notably, D’Hopghe et al. and Fazleabas et al. proposed the seeding and inoculation method to induce peritoneal endometriosis autologously in baboons [49,50]. The induced endometriosis model reveals a similar appearance as spontaneous endometriosis, better yet with higher efficiency [49]. The high expense, strict facilities, and ethical issues still need to be taken into account, which have severely hampered its wide application [51].

#### 3.1.2. Rodents

Rodents such as mice, hamsters, rats, and rabbits have numerous advantages such as cost-effectiveness, small size and large litter size, and the short gestation which enables transgenerational study [51,52,53,54,55]. Furthermore, the wide accessibility of genetically modified mice, abundance of antibodies against rat and murine proteins, and the comprehensive genomic profiles of rodents make them extensively applied to explore the fundamental mechanisms of many diseases, including endometriosis [56]. However, rodents do not get menstruation, which hinders them to develop endometriosis spontaneously. They cannot mimic human conditions as accurately as NHPs. Consequently, the existed rodent models of endometriosis are artificially induced. There are two approaches mostly used to induce endometriosis in rodents: homologous and heterologous. In the homologous model, uteri or the two uterine horns from a donor are implanted in the same or syngeneic recipients. In heterologous models, human endometrium tissues are transplanted into rodents [57,58]. Transplantation by intraperitoneal injection [59] or suturing around the mesentery artery [60] manifests the subtypes of SPE or DIE [61,62,63,64,65]. In rabbits with induced SUP, a decrease in pregnancy rate was reported [61]. A significant reduction in litter size and embryo weights had been observed in the rats model of SUP [62]. It was also indicated that surgically induced SUP in rats resulted in impaired folliculogenesis and oocyte quality [63,64,65]. These models were proved to mimic some characteristics of endometriosis in human cases and provided the insights that endometriotic lesions can negatively affect fecundity.

#### 3.1.3. Organoids

Recent years, organoid, as an in vitro model derived from stem cells, displays the 3D structure and recapitulates biological and pathological features of the original organs [66,67]. They are genetically stable and highly committed to the original tissue during long-term expansion in culture, which enables them to substitute in vitro models, and on the interface of in vitro and in vivo models [67]. Endometrial organoids were firstly cultured by Turco et al. and Boretto et al. in 2017 [68,69], they developed similar characteristics to original tissue, such as hormonal responses during early pregnancy and menstruation [68,69]. For infertility studies, a 3-D organoid model of human endometrium could be developed to study the endometrial interface during embryo implantation [70]. When it was exposed to trophoblast in the implantation window, it showed the process and underlying mechanism of normal or pathological endometrium-embryo crosstalk [70]. For endometriosis research, organoids derived from ectopic endometrium could provide a tool to understand the aetiopathogenesis of endometriosis by comparing the one from normal endometrium [68]. Organoids of endometriosis show the characteristics of endometriosis such as luminal invasion, proliferation, and increased CA-125 level. Combing with genome engineering, it allows genetic intervention and modification for identifying therapeutic targets for further investigation. Besides, the lineage tracing in organoids provides information on cell dynamics, which may be related to the onset, progression, and pathogenesis of endometriosis [71]. Despite the recent development of this technology, there is a lack of studies investigating endometriosis-associated infertility based on organoids.

### 3.2. OMA Models for Infertility

#### 3.2.1. The Current OMA Animal Models

The first description of OMA was in 1899 by Russel. He described the formation of an endometriotic cyst containing glands and uterine epithelium on an ovary [72]. Nearly 100 years later, OMA was first artificially induced in rabbits to explore the effect of OMA on postovulatory events [73]. In this study, the endometrium was separated from the uterus and relocated to one ovary with a fresh incision. Under the same conditions, adipose tissues were transplanted to the contralateral ovary, acting as control. Seven weeks later, nearly 88% of rabbits formed viable implants according to histological analysis [73]. To evaluate the ovulatory function, they counted ovulation points before and after the placement of endometrial or adipose tissues. Results showed a significantly reduced number of ovulation points after endometrial tissue implantation, which indicated that the experimental OMA implied a detrimental effect on the ovulation process [73]. In addition, they graded the periovarian adhesions according to the adhesion density and the proportion of affected ovarian surface. It indicated a higher distribution of extensive adhesions for ovaries with endometrial tissue compared to those with adipose tissue. The ovulation points were significantly decreased in both groups with adhesions, while in the adipose group without adhesions, the number of ovulation points was not affected by the implantation process [73]. This study evaluated the role of periovarian adhesions in OMA and suggested that there might be other factors leading to ovulatory dysfunction in OMA. However, this model and study design remained under some restrictions. Firstly, laparostomy was performed to inspect the ovulation points and periovarian adhesion before implantation. Although the second laparoscopy of tissue placement was arranged three weeks later, the repetitive surgeries in the short term may also be the cause of adhesions. Secondly, the ovulation point was an indirect index for assessing ovulatory function, which was unstable and could impose heterogeneity and subjectivity. With the advancement in modern research, there are lots of tests for ovulation in the clinic, i.e., ovulation hormone test, which can also be applied in animal models for a more accurate result [74]. Besides, there is no further study to explore the subsequent effect on fertility based on this model.

Before 2003, spontaneous OMA was presented as rare in baboons, which was not corresponding to the high frequency in human manifestation, accounting for half of the patients with endometriosis [75,76]. However, a significant prevalence of spontaneous OMA in baboons was reported by Dick et al. in 2003, accounting for 37% of all subjects with endometriosis [77]. The differences in genetics and husbandry environment might contribute to the differences between the two colonies [75,76,78,79]. In Dick’s study, these baboons with OMA had a longer interval between pregnancies, which was considered a potential index of reduced fecundity [77]. This paper demonstrates the feasibility of baboons as a spontaneous model for OMA-associated infertility. Compared to artificially induced animal models, spontaneous models follow a natural routine, which closely resembles human disease. They have the benefits to discover novel molecules associated with traits or pathologies. However, in addition to late manifestation, these models may have uncertain etiology and multifarious causes [80,81].

Most recently, Hayashi et al. established a novel OMA model in mice by attaching uterine tissue around the ovary after the removal of the ovarian bursa which was a membranous structure surrounding the ovaries to protect it from infiltration of ectopic implants [82,83]. Four weeks after implantation, the success rate of the OMA establishment reached 85.7%. Based on this model, the suppressed expression of follicle-stimulating hormone receptor (FSHR) and reduced litter size were found in OMA mice compared with control, indicating that infertility was impaired in OMA mice [82]. However, there were some limitations to this model. Instead of ovaries, implants were also found in nearby tissues and organs, such as the pancreas, muscle, and intestine, manifesting the subtype of SUP and DIE [82]. The involvement of multiple subtypes would cause confounding effects and interact with the specific pathogenesis of OMA-related infertility. Moreover, although the success rate was high, it could not reach 100%, which increased the workload of researchers as histological confirmation of endometriotic lesions was required with this model. Moreover, this model failed to provide the availability of genetic modifications, which could be a strong tool to identify the genetic regulators and therapeutic targets of OMA-related infertility.

Table 1 summaries the current animal models of OMA for studying the related infertility.

#### 3.2.2. The Urge to Establish a Proper OMA Animal Models

Up to now, the current animal models for OMA are quite rare and with slow development. From the above information, it takes more than 10 years to propose a new animal model of OMA. OMA, which act as the most common subtype of endometriosis, may lead to severe dysmenorrhea, infertility, and increases the risk of ovarian cancer [16,84]. While unlike other subtypes of endometriosis, studies on specific OMA models are scarce, which hinders the development of OMA-related studies. It is urgent to develop an appropriate model of OMA for studying its pathology and underlying mechanisms. There is no doubt that these current models of OMA provided us with new insights into the OMA-related infertility, whereas their applications are restrained due to various flaws.

A proper animal model of OMA should be with a promising successful rate, with no ectopic lesion found in surrounding organs except ovaries, and available for genetic modification. In addition, the standard for a proper OMA model should be issued emergently. It is undecided for the type of implants (whole uterus or sole endometrium), transplantation methods, and duration after implantation. According to the ‘3Rs’ principle in animal experiments, it is important to define the suitable implant types, transplantation methods, and duration which might maximize the success rate and minimize the number of animals needed and their suffering time.

## 4. Pathology and Underlying Mechanisms of OMA-Related Infertility Based on OMA Models

### 4.1. Periovarian Adhesions

According to the American Society for Reproductive Medicine (ASRM) scoring, patients with OMA are always staged as moderate or severe as they are usually accompanied by adhesions of surrounding tissues and organs [7,8]. In rabbits with induced OMA, the reduced ovulatory function was significantly correlated with pelvic adhesions at the ovary. Its severity was positively related to decreased ovulation points [73], suggesting that the peri-ovarian adhesions of OMA might impair fecundity. In addition, the surgical treatment for ectopic lesions in the ovary could further facilitate pelvic adhesion and cul-de-sac obliteration, therefore resulting in reduced fertility due to impeded sperm passaging, compromised egg release, and blocked oocyte pickup through fallopian tubes when ovaries embedded within adhesions [38,85].

### 4.2. Ovarian Function

Ovarian reserve reflects ovarian functions from the production of qualified eggs for fertilization, and secretion of ovarian hormones for homeostasis [86]. The reproductive potential comes from dormant primordial follicles activity that is repressed until folliculogenesis takes place in the ovarian cortex [87]. Folliculogenesis is a process that involves follicle activation, development, maturation, and either ovulation of qualified oocyte or follicle atresia. Disturbance of it may lead to oocytes with a reduced number, and poor quality, and subsequently impairs pregnancy outcome [88,89]. Prior clinical studies indicated that the ovarian cortex derived from ovaries with OMA showed a decrease in follicular density compared to that from healthy ovaries [90]. Ovarian response to ovulation stimulation is an indirect indicator of ovarian reserve [91]. Women with unilateral OMA had significantly lower ovarian responsiveness compared to women with contralateral healthy ovaries, demonstrating the presence of OMA lesions may lead to an impaired ovarian reserve [92].

AMH is expressed by GC of developing follicles during folliculogenesis [93]. It prevented primordial follicles from activation and assisted follicle selection for ovulation to maintain ovarian reserve [94]. Serum AMH level is a commonly used marker of ovarian reserve [95]. Its downregulated expression implicated the decline of ovarian reserve, which was found associated with the severity of endometriosis [96]. A study indicated a significantly reduced AMH level in women with previous endometrioma resection, regardless of the current endometrioma lesion [34]. As mentioned before, surgery may reduce ovarian reserve by resecting healthy ovarian tissue [26,28]. Hence, whether it is OMA per se or its resection, or both that damage ovarian reserve is still unclear. As AMH is produced by the preantral and early antral follicles through a dynamic process, its serum level reflects ovarian reserve indirectly [97]. In vivo OMA models provide an opportunity to detect its expression level in ovarian follicles directly. A study on this is still lacking. Follicle-stimulating hormone (FSH) is critical to female fertility via folliculogenesis. The development and maturation of pre-ovulatory follicles are dependent on FSH secretion. FSH is also important for oocyte developmental competence and regulates its subsequent ovulation. It was reported that large follicle survival rate and oocyte quality were significantly decreased in the absence of FSH in vitro [98]. It was also proved that FSH could increase primordial follicle dormancy and ovarian reserve [1]. FSH acts on gonadal target cells by specifically binding to its G protein-coupled receptor, FSHR. The downstream transduction of FSH signaling relies on the receptors [99]. Decreased FSHR distribution leads to a lower response of endogenous FSH, therefore resulting in incomplete reproductive functions [99]. In OMA murine model, FSHR was found to have a dramatically lower expression level in pre-antral, antral, and pre-ovulatory follicles, which might decrease the transduction of FSH signaling reached follicles during folliculogenesis [82]. Less FSH binding to FSHR leads to its accumulation in serum. Thus, a higher serum concentration of FSH was observed to be associated with larger endometriotic lesion size, correlated with lower antral follicle count and oocyte retrieval rate in women with OMA [100].

Ovarian tissues of OMA women were found with a low distribution of primordial follicles, high distribution of growing follicles. It was suggested that premature activation of the ovarian cortex could be the underlying mechanism of declined ovarian reserve caused by OMA [101]. Extensive activation of primordial follicles in a mouse model of peritoneal endometriosis was confirmed to be regulated by the PI3K-PTEN-Akt-Foxo3 signaling pathway [102]. However, the PI3K-PTEN-Akt-Foxo3 pathway was not found significantly different between women with and without OMA [102], yet it was limited by the small sample size of clinical data and the lack of an OMA animal model.

### 4.3. Oxidative Stress

Oxidative stress as a result of excess reactive oxygen species (ROS) and the aberrant antioxidants in the follicular microenvironment, was regarded as essential to induce follicle senescence. This led to disturbed reproductive endocrinology, reduced follicle quality, and density, and ended in subfertility and infertility [103]. More recent studies speculated that excessive oxidative stress and reduced ability of GC might act as causative factors of endometriosis-related infertility via regulating folliculogenesis, oocyte development, ovulation, and embryogenesis [104]. In one study, GCs from infertile patients with or without OMA were collected. Intracellular ROS levels were measured and the result demonstrated increased ROS and excessive oxidative stress in GCs from women with OMA. Their activities impaired ovarian function and were negatively related to oocyte retrieval rate and proportion of mature oocytes [105,106]. The role of oxidative stress in OMA-related infertility had been proved in the murine model of OMA [82]. With endometriotic lesions introduced to the mouse ovaries, fibrosis and iron deposition were found in the endometriotic stroma, representing extensive oxidative stress in the pathogenesis [82]. Though the underlying mechanism of how ROS leads to OMA-related infertility is yet to be understood, it was hypothesized that the toxic contents inside ovarian cyst might induce excessive ROS to increase fibrosis and destruct cortex structure. The blood supply to follicles was reduced, eventually leading to impaired follicle development and decreased follicle count [107]. To prove the hypothesis and explore the potential therapeutic agents that might improve fertility outcomes caused by oxidative stress in OMA, more research should be launched based on OMA models.

## 5. Potential Therapeutics

Therapeutic targets aim at restoring fertility, especially in the aspect of ovarian function are pivotal for the development of novel drugs. Based on the pathogenesis of OMA-related infertility described in part 4, 10 drugs targeting the potential signaling pathways below were screened from the Therapeutic Target Database and listed in Table 2 [108]. These drugs are already available in the market for other diseases, while they could also provide a possibility for their application in the area of OMA-related infertility.

### 5.1. Management of Periovarian Adhesions

Up to now, surgery is the most common management of periovarian adhesions which lead to infertility [109]. Laparoscopy enables aspiration of fresh adhesions, which improves infertility caused by tubal-ovarian adhesion [110]. While there are limited therapeutic targets to remove periovarian adhesions. Some adhesion-reducing substances, i.e., ferric hyaluronic acid, dextran sulfate, and fibrin, decrease post-surgical adhesions only but do not prevent their formation [109]. Furthermore, both surgery and adhesion-reducing substances treatments cannot avoid the recurrence of adhesions [109,110].

### 5.2. Antioxidants

Antioxidants could inactivate, immobilize, or paralyze the ROS before they damage the cells [111]. They can be synthesized endogenously for normal cellular function or derived exogenously to deal with the overabundance of ROS [112]. Quercetin and baicalin, which belong to the flavonoid class, are common-used antioxidants during oocyte development and embryo in vitro maturation [113,114]. As maturation-promoting factor (MPF) and Sirtuins can protect oocytes from aging by decreasing ROS generation, Quercetin delayed oocyte aging after ovulation by attenuating the reduction of MPF activity and regulating the expression of Sirtuins [115]. Mitochondria produce ATP and some ROS, and ROS adversely affects mitochondria function [116]. The abundance of ROS can destroy mitochondria in embryos and thereby decline ATP production, which impairs embryo quality [117]. The Baicalin supplement significantly decreased the ROS level and apoptosis during in vitro embryonic development by improving the mitochondrial membrane potential and ATP level [118]. N-acetyl-cysteine (NAC), an amino thiol with antioxidant properties, prevented damage to bovine embryos when exposed to FF from endometriosis patients with infertility [119]. Melatonin, another antioxidant, is synthesized in the human body naturally. Melatonin regulated against mitochondrial oxidative damage and therefore ameliorated oocyte aging and ovarian reserve exhaustion in mice [108,120,121,122,123,124]. Tafenoquine (TFQ), an FDA-approved antimalarial drug, is applied to treat Trypanosoma brucei infection by targeting ROS and electron transport complex III [125,126]. TFQ showed effectiveness in killing Trypanosoma brucei by increasing ROS production in parasites, thereby leading to programmed cell death by necrosis [127,128]. While TFQ induces cell death by increasing ROS in cells, it should be with limitation in OMA-related infertility as extensive ROS is the possible underlying mechanism to inducing infertility [82].

### 5.3. Restoration of Ovarian Function

MTORC1 and PI3K-PTEN-Akt-Foxo3 pathways were involved in OMA-related infertility [102]. FOXO3, located in the oocyte nucleus of dormant primordial follicles, inhibited activation of primordial follicles and whereby maintained ovarian reserve in the ovarian cortex. FOXO3 phosphorylated through the PI3K-Akt pathway, was transited from nucleus to cytoplasm to initiate primordial follicles activation. The expression level of FOXO3 decreased in ovaries of OMA, leading to premature activation of primordial follicles and subsequent follicle loss. There is no drug reported to increase the FOXO3 level. While a recent study mentioned that melatonin administration inhibited FOXO3 from nuclear transitions [124]. Moreover, FOXO3 is the major target of Akt and can be inactivated by Akt. Melatonin intake was reported to reduce the phosphorylation of Akt, potentially could inhibit Akt activation, and enhanced FOXO3 expression [124]. Akt inhibitor, such as ARQ-751, was also reported to upregulate the expression of FOXO3 [129,130]. Administration of ammonium trichloro (dioxoethylene-o,o’) tellurate (AS101), a modulator of the PI3K-PTEN-Akt pathway, was also observed to prevent hyperactivation of primordial follicles and preserve ovarian reserve in ovarian endometriosis of mice model [102]. There are some drugs (i.e., Sirolimus, Everolimus, Novolimus) commercially available for treating organ transplant rejection, carcinoma, artery stenosis, leukemia, and solid tumor/cancer through regulating mTOR and PI3K-PTEN-Akt pathways [131,132,133,134,135,136,137,138]. Their application and effect on OMA-related infertility could be further investigated.

On the other hand, AMH, which inhibits primordial activation and assists in selective ovulation, was found to lower serum levels in patients with OMA [34,94]. No drug has been reported to improve AMH levels. Vitamin D could significantly increase the serum level of AMH in ovulatory women [139]. Although the correlation between vitamin D and AMH is sophisticated and not fully known, it could be considered a potential nutrient to improve ovarian reserve in infertile women with OMA [139]. GATA binding protein 4 (*GATA-4*) was identified as the regulatory element located in the AMH promoter sequence [139]. It activates AMH transcriptional activity and regulates folliculogenesis of goose [139]. Its overexpression increases AMH expression levels, which may benefit fertility in OMA patients. FSH also plays a central role in regulating the development and maturation of pre-ovulatory follicles, thus helping the folliculogenesis and maintaining mammalian reproduction [140]. Its transduction relies on its receptor, FSHR. There is no drug approved for increasing the FSHR level. Three drugs, Follitropin beta, Menotropins, and Urofollitropin, were recombinant FSH approved for treating female infertility targeting FSHR. While a decreased expression level of FSHR has been reported in mouse models of OMA [82]. The exogenous FSH might have a subtle effect as the signal cannot be transferred through receptors. More investigations on novel drugs are needed for increasing the expression level of FSHR thus improving fecundity in infertile women with OMA.

## 6. Summary and Future Perspective

OMA-related infertility imposes a huge burden on not only individuals and families but also the health care system and society. In order to improve the understanding of fecundity affected by OMA, the establishment of suitable animal models is of great importance to the related studies. Up to now, three animal models of OMA were induced to unmask the key factors responsible for OMA-related infertility, including periovarian adhesion, ovarian dysfunction, and oxidative stress. They constructed a contour about how OMA lesions lead to infertility. In addition, the related molecules and pathways (i.e., PI3K-PTEN-Akt-Foxo3 pathway, excess ROS) provide potential therapeutic targets for OMA-related infertility. It is undisputed that none of these factors can cause OMA-related infertility alonely, and there might be other factors still under investigation.

However, there are limited studies of OMA models available and there is a long-time interval between each study. Except for the one published in 2021, the other two studies were published nearly 20 years ago, which failed to provide trendy perspectives in this area. As well, the lack of further studies based on these two models decreased their conviction.

Moreover, the limitations of current OMA models include the unavailability of genetic modifications or the difficulty to mimic the subtype of OMA hinders the research progress in the pathogenesis of OMA-related infertility. NHP is most proximate to human beings, while its high cost, long period of disease manifestation, difficult access, and maintenance issues prevent it to be the most ideal model of OMA [46,77]. Though the rabbit model of OMA presented a high success rate (88%), it is with a long generation time, long estrus cycle, and is not available for gene modification compared to the mouse model, which impeded its wide application in reproductive diseases [61,73]. An OMA model with a promising success rate, exclusive location of ectopic lesions, and vast availability of genetic modification are desirable. Based on this model, future research can be applied to study fertility indexes, including follicle development, oocyte selection, ovulation, zygote quality, embryo implantation, pregnancy outcomes, etc. Furthermore, a number of on-market drugs which regulate the associated pathways can be repurposed as potential new treatments for OMA-related infertility, followed by adequate preclinical studies and clinical trials that need to be conducted to assess their effectiveness and safety. A proper model can act as an ideal platform for preclinical testing of potential therapeutics, which can accelerate the development of scientific research and novel therapies for OMA-associated infertility.

## Figures and Tables

**Table 1 biomedicines-10-01483-t001:** OMA-associated infertility models summary.

Type	Species/Sources	Year		Method	Fertility Parameters	Limitations	Ref
In vivo	NHP	2003	Baboon	Spontaneously	Total pregnancies	1. Low successful rate of spontaneous model; 2. Long period for disease manifestation; 3. Maintenance issues; 4. Ethical concerns; 5. High cost	[46,77]
	Rodent	1989	Rabbit	Place endometrial tissue in ovaries after incision	Ovulation points	1. Do not develop endometriosis spontaneously 2. Autologous transplantation limited 3. Long generation time; 4. Long estrus cycle 5. Not available for gene modification	[61,73]
		2020	Mouse	Uterine tissue pellet was placed to ovaries after removing ovarian bursa	FSHR, pup numbers	1. Species differences between murine and human; 2. Successful rate of OMA model cannot reach 100%. 3. Lesions was not exclusively located in ovaries.	[82]

NHP: Non-human primates; OMA: endometrioma; FSHR: Follicle-stimulating hormone receptor.

**Table 2 biomedicines-10-01483-t002:** Signaling and potential therapeutic targets explored on OMA models.

Targets	Drug Name ^1^	Approved	Disease
FSHR	Follitropin beta	Y	Female infertility
FSHR	Menotropins	Y	Female infertility
FSHR	Urofollitropin	Y	Female infertility
ROS	Tafenoquine	Y	Plasmodium vivax malaria
Electron transport complex III (Complex III)	Tafenoquine	Y	Plasmodium vivax malaria
PI3K/AKT/mTOR pathway	Sirolimus	Y	Organ transplant rejection
Serine/threonine-protein kinase mTOR (mTOR)	Everolimus	Y	Renal cell carcinoma
Serine/threonine-protein kinase mTOR (mTOR)	Novolimus	Y	Artery stenosis
Serine/threonine-protein kinase mTOR (mTOR)	PF-04449913	Y	Chronic myelomonocytic leukaemia
Serine/threonine-protein kinase mTOR (mTOR)	Sirolimus	Y	Organ transplant rejection
Serine/threonine-protein kinase mTOR (mTOR)	Temsirolimus	Y	Renal cell carcinoma
Serine/threonine-protein kinase mTOR (mTOR)	Zotarolimus	Y	Solid tumour/cancer

^1^ Data were extracted from clinicaltrials.gov.

## Data Availability

Not applicable.

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
