# Peer review of "What We Have Learned from Animal Models to Understand the Etiology and Pathology of Endometrioma-Related Infertility"

_biomedicines, 2022, doi:10.3390/biomedicines10071483_

Round 1
Reviewer 1 Report
In the current review, authors summarized the investigations of OMA-related infertility based on previous and latest OMA models, providing the possible pathogenesis and potential therapeutic targets for further studies. There has been a great load of work here, but I think that some modifications could improve the whole paper and effort of the authors.
Although this is a narrative review, it has to have a more detailed structure, for example authors should add more clear the rationale of this search / reporting of the specific subsections, search strategy, key words, methodology and then results; the final conclusions should be based on the quality of the studies found in the literature during the search; also, the aim of the review should get synchronized with the sections reported afterwards.
There must also be a re post of the figures inside the text.
The final conclusions should be based only in the results of the current search / narrative review and especially on the quality of the studies found. The limitations of the particular study should also been emphasized, leading to the final conclusion and future prospects.
Author Response
In the current review, authors summarized the investigations of OMA-related infertility based on previous and latest OMA models, providing the possible pathogenesis and potential therapeutic targets for further studies. There has been a great load of work here, but I think that some modifications could improve the whole paper and effort of the authors.
Response: Thank you very much for your appreciation and comments. We have made significant modifications, aiming to improve the quality of the manuscript.
Although this is a narrative review, it has to have a more detailed structure, for example authors should add more clear the rationale of this search / reporting of the specific subsections, search strategy, key words, methodology and then results;
Response: Thank you for the valuable suggestions. The manuscript was re-structured to make the rationale clearer in each specific subsection. In section 2 Methodology (line 92-100), we included the search strategy and keywords we used to perform the search of this review. ‘A search was performed on Pubmed using the search terms (("Endometriosis"[Mesh]) AND (("Fertility"[Mesh]) OR ("Infertility"[Mesh]) OR ("Reproduction"[Mesh]))) AND ("Models, Animal"[Mesh]) on 05/02/2022. All titles and abstracts were screened to verify the mentioned animal model of endometriosis-related infertility and 128 articles were screened from the beginning. 39 related articles were extracted from other reviews. After screening out reviews, irrelevant and duplicate articles, 82 articles were identified. Thereinto, 3 articles indicated the animal models for OMA in endometriosis-associated infertility’.
the final conclusions should be based on the quality of the studies found in the literature during the search;
Response: Thank you for the valuable suggestions. In lines 455-459, we added the conclusion based on the quality of studies, pointed out their limitations, and advocated for more research in this area.
also, the aim of the review should get synchronized with the sections reported afterwards.
Response: Thank you for pointing this out. We made a revision to synchronize the aim of the review, which is focused on animal models and pointed out the urgent to establish a more appropriate animal model of OMA to study its related infertility. The related statements can be found in lines 249-266, 467-469.
There must also be a re post of the figures inside the text.
Response: Thank you for pointing this out, we have rearranged the figures and tables inside the text. The graphic abstract can be found in line 23. Table 1 can be found in line 267, and table II can be found in line 366.
The final conclusions should be based only in the results of the current search / narrative review and especially on the quality of the studies found. The limitations of the particular study should also been emphasized, leading to the final conclusion and future prospects.
Response: Thank you for the valuable suggestions. In the part of ‘Summary and Future perspective’, we added the conclusion based on the quality of studies, pointed out their limitations, and advocated for more research in this area. We added the following sentences in lines 455-459; 460-462; 467-469. ‘However, there are limited studies of OMA models available and there is a long-time interval between each study. Except the one published in 2021, the other two studies were published nearly 20 years ago, which failed to provide trendy perspectives in this area. And the lack of further studies based on these two models decreased their conviction’; ‘Moreover, the limitations of current OMA models include the unavailability of genetic modifications or the difficulty to mimic the subtype of OMA hinders the research progress in the pathogenesis of OMA-related infertility’; ‘An OMA model with a promising success rate, exclusive location of ectopic lesions, and vast availability of genetic modification is desirable.’

Reviewer 2 Report
Tan et al review ovarian endometriosis (endometrioma) and its impact on fertility. Major work is needed on this manuscript before publication, as detailed below:
Major problem
The paper is far too long, mainly due to frequent diversions beyond the topic advertised in the title, which is animal models. If the authors wish to limit their review in this way, then they are obliged to cull out any material which does not pertain to veterinary research.
Minor issues
Lines 15,35,36 etc: Define subtype. Is this different from type? If not, why use an invented taxonomy?
Line 16-17: “coupled with undesired infertility … “ This expression is both awkward and probably incorrect. Ovarian cysts are often painful, irrespective of fertility implications. Surely there is recognition that even mild pelvic endometriosis is linked with severe dysmenorrhea and/or dyspareunia?
Line 22: summarized (why use past tense here?)
Line 36: check syntax
Lines 41,42: “thereinto” & “most essential manifestations” are these really the best word choices?
Line 54: “primary” is unnecessary
Beginning at Line 55”: Consider deleting this confusing section entirely. The authors correctly list several Rx meds used to treat endometriosis, but then comment these “under the mechanism of ovulation suppression and do not benefit fertility” (line 58) and “may not be suitable” if fertility is desired. These are all established first-line agents which share the chief aim of reducing endogenous estrogen effects. Blocking ovulation is just a secondary consequence of this action. Indeed OCPs and GnRH-analogs are well-known components of programmed IVF stims.
“Medicals” means medications? (line 59)
Line 60: Can robotic surgery be offered independent of laparoscopy? Introducing the robot-assisted surgery topic will attract criticism on a separate front, as this technology in GYN surgery has never been shown to be cost-effective. To claim this is “widely used” is quite misleading, and surely needs a citation.
Line 73 etc: The EFI has dubious value in clinical IVF practice and few ART programs probably even know it exists. No wonder the reference given is from 2010, as it has largely fallen into disuse since then.
Figure 1. This roadmap could be omitted and nothing instructive is lost.
Line 375-7 & 451-2: These do not appear to be complete sentences.
Line 408 etc. Getting into the subject of epigenetics is notable, yet the only papers mentioned here are on murine ROS or other work from nearly 20yrs ago. Recommend updating the review, or simply deleting this section if more current research cannot be found to support the narrative.
Line 478 etc. Unsure what is gained by including TNF inhibitors, TFQ, T. brucei etc. all of which are extremely tangential to endometriosis. There are limitless irrelevant or futile areas of medical investigation which could draw notice, but don't warrant specific comment.
Line 521 etc. A discussion on IVF stim protocols is a distraction to this review. Note there is disagreement on ‘best gonadotropin regime’ for unselected IVF patients, and this variation applies to endometriosis cases, too. Coverage of IVF drugs settles nothing here. The matter is sufficiently complex on its own to deserve its own review, rather than being inappropriately included in this paper - which is supposed to be on animals.
Line 559: organoid?
Line 575: please define/explain what is meant by repurposing “on-market drugs” which regulate pathways. If these have contraindications akin to those medications mentioned at line 55 etc, then why even mention such remote hypotheticals?
Any referenced paper dated more than 10yrs ago only shows how minimally a field has advanced.
Why put ‘past, present, and future’ in this title - what else is there?
Author Response
Tan et al review ovarian endometriosis (endometrioma) and its impact on fertility. Major work is needed on this manuscript before publication, as detailed below:
Major problem
The paper is far too long, mainly due to frequent diversions beyond the topic advertised in the title, which is animal models. If the authors wish to limit their review in this way, then they are obliged to cull out any material which does not pertain to veterinary research.
Response: Thank you for the valuable suggestions. We appreciate the time and effort that you have dedicated to provide the valuable comments and suggestions. We changed the title from ‘Insights from animal models of endometrioma in disease modeling and therapeutic applications for infertility: the past, present, and future’ to 'What we have learned from animal models to understand the etiology and pathology of endometrioma-related infertility’ to focus more on the current animal models of OMA, their application to reveal the etiology and pathology of OMA-related infertility, and their limitation that leads to the needs of a proper OMA model for further study. We made significant revisions to the manuscript and cut down the word counts from 6277 to 5354. In order to focus on the animal model of OMA, we deleted the sections on in vitro models and the pathogenesis and potential therapeutic targets obtained from these models. Our revised manuscript mainly focused on animal models of OMA. Therefore, we made significant changes especially in the part 3 ‘Animal models of endometriosis-associated infertility and OMA-related infertility’, detailed pros and cons of each animal model, as well as their feasibility and limitation to be translatable to the clinic, which were also listed out as below:
- In ‘3.1.1 non-human primates’, we introduced NHPs and the pros (lines 124-129) and cons (lines 134-142) of its application, as well as their similarity to human endometriosis (lines 129-130) and transferability to evaluate the disease progression and monitor drug efficacy in preclinical trials (lines 130-133).
- In ‘3.1.2 Rodents’, we summarized the advantages of rodent models (line 145-150), their shortcomings (lines 150-152), and the similar reflection of this model to humans in endometriosis-associated infertility (lines 159-164).
- In ‘3.2.1 The current OMA animal models’, we described the methods of each OMA model establishment. The first method artificially induced OMA in rabbits by placing endometrium in ovaries with a fresh cut (lines 194-197). Its reflection of human conditions was described (lines 199-209), and the limitations of this method was listed in line 209-218. The second OMA model was induced spontaneously in NHPs (Baboons). The reflection of baboon models of OMA to human conditions is described in lines 224-227. And its advantages and disadvantages are delineated in lines 227-231. Thirdly, OMA mouse model was artificially induced by attaching uterine tissue around the ovary after the removal of the ovarian bursa (lines 232-236). The application of OMA mouse model was represented in lines 236-238. Its limitations are also mentioned in lines 238-247.
In ‘3.2.2 The urge to establish a proper OMA animal model”, we narrated the emergency of establishing OMA animal models (line 250-258) and provided criteria (line 259-261) and standard (261-266) for a proper OMA animal model.
Minor issues
Lines 15,35,36 etc: Define subtype. Is this different from type? If not, why use an invented taxonomy?
Response: Thank you. We defined the subtypes of endometriosis in lines 26-28, 29-32. Endometrioma (OMA) occurs within ovaries and manifests as single or multiple distinct cysts. Superficial peritoneal endometriosis (SUP) lies on the lining of the peritoneum and deep infiltrating endometriosis (DIE) is characterized by lesions infiltrating over 5mm under the peritoneal surface. OMA, SUP, and DIE are recognized as the major phenotypes of endometriosis and are the commonly used taxonomy.
Line 16-17: “coupled with undesired infertility … “This expression is both awkward and probably incorrect. Ovarian cysts are often painful, irrespective of fertility implications. Surely there is recognition that even mild pelvic endometriosis is linked with severe dysmenorrhea and/or dyspareunia?
Response: Thank you for pointing this out and sorry for the confusion caused. We amended the sentence “coupled with undesired infertility’’ to ‘associated with infertility’ (line 15). We aim to clearly emphasize the correlation between endometriosis and infertility, which had been demonstrated in a lot of human studies. As endometriosis is likely to happen in women of reproductive age, the related infertility may lead to a huge burden to those who have conceive desire. A high frequency of endometriosis-related infertility has been reported by the Practice Committee of the American Society for Reproductive Medicine [1]. It is delineated that 30% to 50% of patients with endometriosis are infertile, and among women with infertility, the prevalence of endometriosis reached up to 50% [2]. While the cost in the reproductive centre is quite high, one single IVF cycle can range from $15,000 to $30,000 [3], which proposes considerable economic stress to those families. We also agreed that the symptoms of severe pain are common in patients with endometriosis including OMA and other subtypes. However, several possible causes may contribute to the associated pain depending on different subtypes of endometriosis [4], which is complicated and should be comprehensively summarized for another review.
Line 22: summarized (why use past tense here?)
Response: Thank you for pointing this out. We changed it to present tense.
Line 36: check syntax
Response: Thank you. We corrected the sentence to ‘There are other two subtypes of endometriosis: 1) superficial peritoneal endometriosis (SUP) and 2) deep infiltrating endometriosis (DIE). The former one lies on the lining of the peritoneum and the latter one is characterized by lesions infiltrating over 5mm under the peritoneal surface [5].’ (line 29-32)
Lines 41,42: “thereinto” & “most essential manifestations” are these really the best word choices?
Response: Thank you. We deleted the statement and emphasized the manifestations and stages of OMA (line 35-36)
Line 54: “primary” is unnecessary.
Response: Thank you. We deleted ‘primary’ in this sentence.
Beginning at Line 55”: Consider deleting this confusing section entirely. The authors correctly list several Rx meds used to treat endometriosis, but then comment these “under the mechanism of ovulation suppression and do not benefit fertility” (line 58) and “may not be suitable” if fertility is desired. These are all established first-line agents which share the chief aim of reducing endogenous estrogen effects. Blocking ovulation is just a secondary consequence of this action. Indeed OCPs and GnRH-analogs are well-known components of programmed IVF stims.
Response: Thank you for the valuable suggestions and sorry for the confusion caused. We made revisions and included current therapeutic options for endometriosis, as well as briefly stated their mechanism of action (line 48-56). First-line agents of current medication of endometriosis include combined oral contraceptives (COC) and progestin, which reduce estrogen effects and suppress ovulation. These drugs significantly reduced the recurrence of dysmenorrhea [6]. While their application leads to the risk of impaired infertility, symptoms related to hypoestrogenism, and thromboembolism. Gonadotropin-releasing hormone (GnRH) agonist is the second-line therapy of endometriosis to ameliorate the symptoms and prevent recurrence after surgery by suppressing ovarian production of estrogen and creating a hypoestrogenic state. Similarly, it also brings long-term side effects such as osteoporosis because of the increased bone turnover and subsequent reduction in bone mineral density [7, 8]. Surgical treatments are widely used for the diagnosis or lesion resection [9]. However, it is still under debate whether surgical treatment of OMA removal will benefit pregnancy outcomes. The ovarian reserve assessed by antral follicle count (AFC) was not reduced with surgical removal of OMA in one meta-analysis, while a systematic review and prospective cohort studies reported a decreased ovarian reserve as the consequence of surgical excision of OMA [10-12]. The discrepancy may come from variations in surgical skills between surgeons and different stages of endometriotic lesions. Instead of the possibility of ovarian injury [13], surgical interventions may lead to other potential side effects, such as postoperative adhesions, surgical complications, delayed infertility treatment, and high recurrence rate [14-15]. Artificial reproduction technology (ART) was also proposed to apply solely or combined with surgery [14-17]. While it probably has subtle benefits to women with OMA. A systematic review reported that during solely application of ART, patients with OMA had a higher cycle cancellation rate and lower oocyte yield [18]. According to ESHRE guideline, there was no evidence to show that the surgical removal of OMA lesion before ART could improve the pregnancy rate. Overall, in this section (lines 46-79), we aimed to provide the restriction of current therapies for OMA-related infertility, emphasize the importance of improving the current therapies and investigating novel therapeutic targets for the development of new therapies.
“Medicals” means medications? (line 59)
Response: Thank you. We changed it to 'medications’.
Line 60: Can robotic surgery be offered independent of laparoscopy? Introducing the robot-assisted surgery topic will attract criticism on a separate front, as this technology in GYN surgery has never been shown to be cost-effective. To claim this is “widely used” is quite misleading, and surely needs a citation.
Response: Thank you. Although it was mentioned in gynecology diseases, robot-assisted surgery can be highly advantageous [19], we agreed that introducing the robot-assisted surgery topic would attract criticism, and therefore we deleted this section.
Line 73 etc: The EFI has dubious value in clinical IVF practice and few ART programs probably even know it exists. No wonder the reference given is from 2010, as it has largely fallen into disuse since then.
Response: Thank you for pointing this out. We agreed that EFI has dubious value in clinical IVF practice and as it has largely fallen into disuse since 2010. Therefore, we deleted the related sentences.
Figure 1. This roadmap could be omitted and nothing instructive is lost.
Response: Thank you very much for your suggestion. The roadmap has been deleted.
Line 375-7 & 451-2: These do not appear to be complete sentences.
Response: Thank you for pointing this out. We revised the sentence and changed it to ‘It was suggested that premature activation of the ovarian cortex could be the underlying mechanism of declined ovarian reserve caused by OMA’ (line 325-327).
Line 408 etc. Getting into the subject of epigenetics is notable, yet the only papers mentioned here are on murine ROS or other work from nearly 20yrs ago. Recommend updating the review, or simply deleting this section if more current research cannot be found to support the narrative.
Response: Thank you for the valuable suggestions. In order to focus on the animal model of OMA, we deleted the sections on in vitro models.
Line 478 etc. Unsure what is gained by including TNF inhibitors, TFQ, T. brucei etc. all of which are extremely tangential to endometriosis. There are limitless irrelevant or futile areas of medical investigation which could draw notice, but don't warrant specific comment.
Response: Thank you for pointing this out. We have depleted this part.
Line 521 etc. A discussion on IVF stim protocols is a distraction to this review. Note there is disagreement on ‘best gonadotropin regime’ for unselected IVF patients, and this variation applies to endometriosis cases, too. Coverage of IVF drugs settles nothing here. The matter is sufficiently complex on its own to deserve its own review, rather than being inappropriately included in this paper - which is supposed to be on animals.
Response: Thank you for the valuable suggestions and sorry for the distraction. We deleted the ovarian stimulation protocols in IVF-ET.
Line 559: organoid?
Response: Thank you for pointing this out. To avoid misunderstanding, we deleted organoid in this part.
Line 575: please define/explain what is meant by repurposing “on-market drugs” which regulate pathways. If these have contraindications akin to those medications mentioned at line 55 etc, then why even mention such remote hypotheticals?
Response: Thank you. Firstly, we have changed the subtitle of part 5 to ‘Potential therapeutics. These drugs are targeting the molecules and potential signaling pathways described in part 4. We defined ‘on-market drugs’ as drugs that have accomplished the clinical trials and passed safety tests in other disease. By repurposing these drugs and to apply in the treatment of OMA-related infertility, they are with higher availability and feasibility, compared to drugs in investigating stages. Secondly, compared to the current medications of endometriosis, they may inhibit endometriosis with different mechanisms. The potential drugs listed in section 5 (lines 357-442) target the molecules and signaling pathways that potentially could folliculogenesis, improve oocyte quantity and quality, and manage excessive ROS based on the discussed pathology and underlying mechanisms of OMA-related infertility in section 4. Therefore, the application of these drugs might with potential improve fecundity in patients with OMA. However, more experimental and clinical studies should be launched.
Any referenced paper dated more than 10yrs ago only shows how minimally a field has advanced.
Why put ‘past, present, and future’ in this title - what else is there?
Response: Thank you. We made a revision in the title and changed it to ‘What have learned from animal models to understand the etiology and pathology of endometrioma-related infertility’. Firstly, we described the current animal models of OMA and how they were applied to study the etiology and pathology of related infertility. And then we mentioned the shortage and limitations of current animal models of OMA. Up to now, only three animal models of OMA have been established. Among these three models, two were introduced from literature published 20 years ago. However, OMA-related infertility is always an important manifestation that impose a huge burden not only to the individual but also the society. The lack of proper animal models impedes the related study progress in this area. Our current review not only summarized the current OMA animal models but also aimed to encourage researchers to build animal models of OMA for their further applications in the pathogenesis and novel therapeutics. For this reason, we also suggested the criteria and standard of appropriate animal models of OMA in part 3.2.2

Round 2
Reviewer 2 Report
The work now has been improved in response to prior comments and may be considered for publication, if the Chief Editor agrees.